# RAPID: Long-Context Inference with Retrieval-Augmented Speculative Decoding

**Guanzheng Chen** [* 1 2 3] **Qilong Feng** [* 1] **Jinjie Ni** [1] **Xin Li** [2 3] **Michael Qizhe Shieh** [1]

Code: https://github.com/NUS-TRAIL/RAPID

## Abstract

The emergence of long-context large language models (LLMs) offers a promising alternative to traditional retrieval-augmented generation (RAG) for processing extensive documents. However, the computational overhead of long-context inference presents significant efficiency challenges. While Speculative Decoding (SD) traditionally accelerates inference using smaller draft models, its effectiveness diminishes substantially in long-context scenarios due to memory-bound KV cache operations. We introduce **R**etrieval-**A**ugmented S**P**eculat**I**ve **D**ecoding (**RAPID**), which leverages RAG for both accelerating and enhancing generation quality in long-context inference. RAPID introduces the RAG drafter—a draft LLM operating on shortened retrieval contexts—to speculate on the generation of long-context target LLMs. Our approach enables a new paradigm where same-scale or even larger LLMs can serve as RAG drafters while maintaining computational efficiency. To fully leverage the potentially superior capabilities from stronger RAG drafters, we develop an inference-time knowledge transfer that enriches the target distribution by RAG. Extensive experiments on the LLaMA-3.1 and Qwen2.5 backbones demonstrate that RAPID effectively integrates the strengths of both RAG and long-context LLMs, achieving significant performance improvements (e.g., from 39.33 to 42.83 on InfiniteBench for LLaMA-3.1-8B) with more than $2\times$ speedups for long-context inference. Our analyses also reveal the robustness of RAPID across various context lengths and retrieval quality.

---

[*]Equal contribution [1]National University of Singapore [2]DAMO Academy, Alibaba Group [3]Hupan Lab, 310023, Hangzhou, China. Correspondence to: Guanzheng Chen <gc.chen@u.nus.edu>, Michael Qizhe Shieh <michaelshieh@comp.nus.edu.sg>.

*Proceedings of the $42^{nd}$ International Conference on Machine Learning*, Vancouver, Canada. PMLR 267, 2025. Copyright 2025 by the author(s).

## 1. Introduction

Large language models (LLMs) have traditionally relied on retrieval-augmented generation (RAG) to process extensive documents by selectively retrieving relevant text segments. While effective, the performance of RAG is inherently bounded by the capability of the retriever to extract pertinent information across diverse queries (Gao et al., 2023). The recent emergence of long-context LLMs, capable of directly processing million-word documents (Team et al., 2024), suggests a promising alternative to complex RAG pipelines. However, this breakthrough is bottlenecked by the computational efficiency of long-context inference, where processing extensive key-value (KV) caches becomes memory-bound and introduces substantial latency (Pope et al., 2022).

Speculative Decoding (SD) (Chen et al., 2023; Leviathan et al., 2023) is a prevalent approach to accelerate LLM inference without compromising generation quality. By leveraging a smaller draft model to propose multiple candidates for single-pass validation by the target model, SD achieves significant speedup when candidates are accepted. The benefits of SD hinge on two critical factors: the computational efficiency of the draft model in generating candidates, as well as its capability to produce high-quality and acceptable candidates. However, SD will become less effective in long-context scenarios, as memory-bound KV cache operations prevent smaller LLMs from maintaining significant speed benefits over larger models (Pope et al., 2022; Ainslie et al., 2023b). As depicted in Figure 1, the throughput gains of LLaMA-3.1-8B over LLaMA-3.1-70B diminish drastically ($23.6 \rightarrow 9.4$) with increasing context lengths from 1K to 128K tokens.

In this work, we introduce **R**etrieval-**A**ugmented S**P**eculat**I**ve **D**ecoding (**RAPID**), to bridge the gap of SD for accelerating long-context inference while enhancing generation quality. RAPID employs a *RAG drafter*—the draft LLM operating on shortened context from RAG—to speculate the generation of long-context LLM following the SD process. We propose that RAG drafter can serve as *ideal draft model* for long-context target LLM, as it

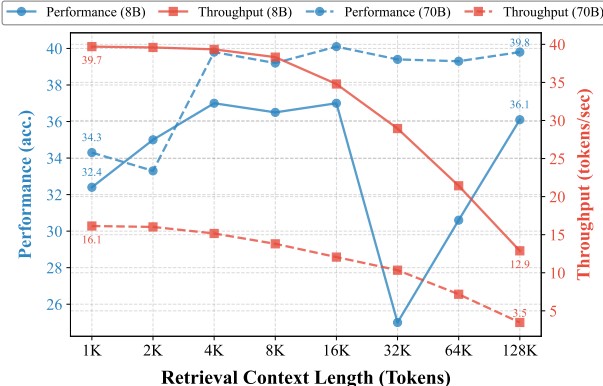

*Figure 1.* Performance (accuracy, left axis) and throughput (tokens/sec, right axis) of LLaMA-3.1-8B (serving on 1×A800) and LLaMA-3.1-70B (serving on 8×A800) on LongBench v2 (Long) across different retrieval context lengths.

demonstrates the potential to approach the capabilities of long-context LLM (Li et al., 2024b) while offering superior computational efficiency. As illustrated in Figure 1, LLaMA-3.1-8B with RAG on 4K∼16K tokens can recover most performance achieved with full 128K tokens. This indicates that the RAG drafter is capable of producing high-quality candidates for long-context target LLM with high acceptance rate, while eliminating the memory-bound KV cache operations over long-context to accelerate the inference process.

In addition, our RAPID opens a new paradigm for SD that *leveraging the same-scale or even larger LLMs as the RAG drafters to accelerate smaller target LLMs.* This paradigm shift is possible since RAG drafters, operating on shortened contexts (e.g., 4K), potentially maintain higher efficiency than target LLMs of the same or even larger scale on long contexts (e.g., 128K) as evidenced in Figure 1. Therefore, our RAPID operates on two settings: (1) *self-speculation*, where long-context target LLM and RAG drafter are of the same scale; and (2) *upward-speculation*, where RAG drafter involves larger parameter scale than target LLM. Moreover, in the both settings, the generation quality of RAG drafter may surpass that of long-context target models in some scenarios (Li et al., 2024a). However, the native SD, utilizing target LLM prediction as ground-truth distribution to perform rejection sampling, may neglect the candidates of high quality from the stronger RAG drafter. This would result in unnecessary rejection of valid candidates, thereby impeding both efficiency and performance gains.

To address this limitation, RAPID implements a retrieval-augmented target distribution, which incorporates the native long-context target distribution in SD with an *inference-time knowledge transfer*. Specifically, we reversely position the RAG drafter as teacher and long-context target LLM as the student, to derive a distilled logits shift towards the RAG drafter during inference. By incorporating the shift into the prediction logits of target LLM, we obtain an enriched target distribution that is more receptive to high-quality speculative candidates.

Our RAPID can serve as a drop-in decoding method during long-context inference. We conduct experiments on LLaMA-3.1 (8B, 70B) (Dubey et al., 2024) and Qwen2.5 (7B, 72B) (Yang et al., 2024) series on ∞Bench (Zhang et al.) and LongBench v2 (Bai et al., 2024b). The experimental results demonstrate that RAPID successfully integrates the complementary strengths of long-context LLMs and RAG while maintaining significant inference speedups. In self-speculation settings, RAPID achieves consistent performance improvements (e.g., 42.83 vs 39.33 on InfiniteBench for LLaMA-3.1-8B) with significant speedup (up to 2.69×) over the long-context target LLMs. The upward-speculation setting further boosts performance through effective knowledge transfer from larger RAG drafters (e.g., improving LLaMA-3.1-8B from 42.83 to 49.98 on InfiniteBench), with comparable efficiency with the smaller long-context target LLMs. With moderate retrieval length (≤16K) for RAG drafter, we found RAPID consistently achieves speedup when target long-context length beyond 32K. Our analyses also indicate that RAPID demonstrates robustness to retrieval quality and potentially superior generation quality in real-world multi-turn dialogue tasks. These results validate RAPID as an effective decoding method for accelerating long-context inference and, at the same time, enhancing generation quality through retrieval-augmented speculation.

## 2. RAPID: Retrieval-Augmented Speculative Decoding

### 2.1. Background: Speculative Decoding

Autoregressive generation with a LLM $p_\phi$ traditionally requires sequential forward passes, where each token $x_i$ is sampled from the distribution $p_\phi(x_i|x_{<i})$. This sequential nature incurs substantial computational overhead for LLM parameters loading and KV cache manipulation in GPU DRAM. SD accelerates this process using a smaller draft model $q_\psi$ to generate $\gamma$ candidate tokens, which are then validated by the target model $p_\phi$ in a single forward pass through rejection sampling. For each speculative token $x_i' \sim q_\psi(x_i|x_{<i})$, the acceptance criterion is:

$$r \le \min\left(1, \frac{p_\phi(x_i'|x_{<i})}{q_\psi(x_i'|x_{<i})}\right), \quad (1)$$

where $r \sim U(0, 1)$. Upon rejection, a new token is sampled from the residual distribution:

$$x_i \sim \texttt{norm}(\max(p_\phi(x_i|x_{<i}) - q_\psi(x_i|x_{<i}), 0)), \quad (2)$$

where `norm` is to normalize the distribution by $\ell_1$ norm.

This procedure guarantees that the resampled tokens follow the exact distribution as direct sampling from the target model $p_\phi$, while potentially achieving significant speedup when the speculative tokens are accepted.

## 2.2. Overview

While traditional SD offers significant speedups for standard-length contexts, its benefits diminish substantially when handling extensive documents due to memory-bound KV cache operations. We present RAPID, a method that reimagines SD for long-context scenarios while enhancing generation quality. As demonstrated in Alg. 1, RAPID comprises two critical components:

**RAG Drafter.** SD becomes inefficient with long contexts as both draft and target LLMs must process complete context in memory, negating the computational advantages of smaller drafter. To overcome this challenge, RAPID utilizes a *RAG drafter* to generate candidates for long-context LLMs as introduced in §2.3. The RAG drafter operates on selectively retrieved context segments, enabling significant speedups while maintaining access to relevant information.

**Retrieval-Augmented Target Distribution.** The strict acceptance criterion in SD may reject high-quality candidates, as it requires strict match to the target LLM distribution for acceptance. This constraint becomes particularly limiting when using RAG drafters, which can potentially generate higher-quality outputs than long-context LLMs in certain scenarios (Li et al., 2024a). To incorporate the benefits from RAG drafters, RAPID steers a retrieval-augmented target distribution (§2.4), which enables knowledge transfer from RAG drafter to target model during inference. This mechanism allows the target distribution to incorporate valuable information while maintaining theoretical guarantees of the original SD.

## 2.3. RAG Drafter

When processing queries for extensive context $\mathcal{C}$, the target distribution of naive SD is

$$p(x_i) = p_\phi(x_i|[\mathcal{C}; x_{<i}]). \tag{3}$$

Even with smaller draft models, the computational benefits diminish substantially due to memory-bound KV cache operations over the complete context $\mathcal{C}$. To overcome this limitation, we propose to leverage RAG as the foundation for our draft model.

Instead of processing the entire context $\mathcal{C}$, our RAG drafter operates on a compressed context $\mathcal{C}^S$. Specifically, $\mathcal{C}^S$ is constructed through selective retrieval: text segments from $\mathcal{C}$ are encoded into a dense vector space, where semantic similarity to the query is measured via cosine similarity,

---

**Algorithm 1** Retrieval-Augmented Speculative Decoding

**Require:** Target LLM $p_\phi$, RAG drafter $q_\psi$, context $\mathcal{C}$, retrieval context $\mathcal{C}^S$, number of speculative tokens $\gamma$, temperature $T$, transfer strength $\eta$

**Ensure:** Generated sequence $x_{1:n}$

1: $i \leftarrow 1$
2: **while** $i \leq n$ **do**
3:    // Generate $\gamma$ speculative tokens using RAG drafter
4:    **for** $k \leftarrow 1$ to $\gamma$ **do**
5:      $x'_{i+k-1} \sim q(x_{i+k-1}) = q_\psi(\cdot|[\mathcal{C}^S; x_{<i}; x'_{i:i+k-1}])$
6:    **end for**
7:    // Validate speculative tokens sequentially
8:    **for** $k \leftarrow 1$ to $\gamma$ **do**
9:      $j \leftarrow i + k - 1$
10:     // Compute target and draft distributions
11:     $z(x'_j) \leftarrow \texttt{LogitsOf}(p_\phi(\cdot|[\mathcal{C}; x_{<j}]))$
12:     $p(x'_j) \leftarrow \text{softmax}(z(x'_j)/T)$
13:     $q(x'_j) \leftarrow q_\psi(x'_j|[\mathcal{C}^S; x_{<j}])$
14:     // Compute retrieval-augmented target distribution
15:     $\hat{z}(x'_j) \leftarrow z(x'_j) + \eta T(q(x'_j) - p(x'_j))$ (Eq. (8))
16:     $\hat{p}(x'_j) \leftarrow \text{softmax}(\hat{z}(x'_j)/T)$
17:     $r \sim U(0, 1)$
18:     **if** $r \leq \min(1, \frac{\hat{p}(x'_j)}{q(x'_j)})$ **then**
19:       $x_j \leftarrow x'_j$
20:       $i \leftarrow j + 1$
21:     **else**
22:       **goto** line 26
23:     **end if**
24:    **end for**
25:    // Sample from residual if rejected
26:    $x_i \sim \texttt{norm}(\max(p(x_i) - \hat{p}(x_i), p(x_i) - q(x_i), 0))$
27:    $i \leftarrow i + 1$
28: **end while**
29: **return** $x_{1:n}$

---

enabling efficient identification and extraction of the most relevant context chunks.

After deriving the compress context $\mathcal{C}^S$, the draft distribution is formally defined as

$$q(x_i) = q_\psi(x_i|[\mathcal{C}^S; x_{<i}]), \tag{4}$$

where we maintain strict control over the compression ratio by enforcing $|\mathcal{C}^S| \leq |\mathcal{C}|/\lambda$ with $\lambda \gg 1$. This compressed context enables our draft model to maintain significant speed advantages while preserving access to relevant information.

Based on the RAG drafter, the modified speculative decoding process proceeds as follows. For each generation step, we sample $\gamma$ speculative tokens from the RAG drafter as $x'_i \sim q(x_i)$. These candidates are validated against the target

model using a modified acceptance criterion:

$$r \leq \min\left(1, \frac{p(x_i)}{q(x_i)}\right) = \min\left(1, \frac{p_\phi(x_i'|[\mathcal{C}; x_{<i}])}{q_\psi(x_i'|[\mathcal{C}^S; x_{<i}])}\right) \quad (5)$$

where $r \sim U(0, 1)$.

The RAG-based drafting mechanism offers two key advantages: (1) significant reduction in memory overhead and computational cost through compressed context operations ($|\mathcal{C}^S| \ll |\mathcal{C}|$), and (2) potentially enhanced speculation quality through selective retrieval of relevant information compared to processing diluted full context. Moreover, due to the remarkable efficiency on shorten context, RAPID even enables the use of same-scale or larger models as drafters to accelerate smaller target LLMs.

### 2.4. Retrieval-Augmented Target Distribution

The capability of LLMs to effectively utilize context often deteriorates with irrelevant information inclusion. Our empirical analysis in Figure 1 shows that LLMs, by focusing on retrieved relevant chunks, can sometimes surpass full-context utilization in generation quality. However, the strict acceptance criterion of traditional SD may potentially result in unnecessary rejection for these superior generations when they deviate from the target distribution, leading to both quality degradation and computational inefficiency.

To address this limitation, we introduce retrieval-augmented target distribution, which enables knowledge transfer from the RAG drafter to the long-context target model during inference. Formally, the retrieval-augmented target distribution in RAPID is defined as:

$$\hat{p}(x_i) = \text{softmax}(z(x_i)/T + \eta \cdot (q(x_i) - p(x_i))), \quad (6)$$

where $\eta$ is a hyperparameter controlling the strength of knowledge transfer, $z(x_i)$ is the unnormalized logits of target LLM, namely $p(x_i) = \text{softmax}(z(x_i)/T)$ and $T$ is the temperature.

**Proposition 2.1.** *Let* $p(x) = \text{softmax}(z(x)/T)$ *be a student model distribution parameterized by logits* $z(x)$ *and temperature* $T$, *and* $q(x)$ *be a teacher model distribution. The gradient of the knowledge distillation loss* $\mathcal{L} = T^2 \cdot KL(q(x)\|p(x))$ *with respect to* $z(x)$ *is:*

$$\frac{\partial \mathcal{L}}{\partial z(x)} = T \cdot (p(x) - q(x))$$

*where* $KL(\cdot\|\cdot)$ *denotes the Kullback-Leibler divergence.*

*Proof.* See Appx. §A. □

The design of retrieval-augmented target distribution in Eq. (6) implies a knowledge distillation step by positioning the RAG drafter as the teacher and the target model

as the student, to infuse a proportion of knowledge from RAG drafter into naive long-context target distribution.

Specifically, for a distillation loss (Hinton et al., 2015) $\mathcal{L}$ between RAG draft distribution $q(x_i)$ (teacher) and long-context target distribution $p(x_i)$ (student), according to Proposition 2.1, we have the distilled logits shift as

$$\frac{\partial \mathcal{L}}{\partial z(x_i)} = T \cdot (p(x_i) - q(x_i)). \quad (7)$$

Now we can derive a "distilled" $z(x_i)$ augmented by RAG drafter through

$$\begin{aligned} \hat{z}(x_i) &= z(x_i) - \eta \frac{\partial \mathcal{L}}{\partial z(x_i)} \\ &= z(x_i) + \eta T(q(x_i) - p(x_i)), \end{aligned} \quad (8)$$

where $\eta$ controls the strength of knowledge transfer. Therefore, the retrieval-augmented target distribution in Eq. (6) is equivalent to the normalized $\hat{z}(x_i)$, i.e., $\hat{p}(x_i) = \text{softmax}(\hat{z}(x_i)/T)$.

The retrieval-augmented target distribution $\hat{p}(x_i)$ enables flexible knowledge transfer from the RAG drafter while maintaining verification capability. Since the unnormalized logits $z(x_i) \in \mathbb{R}$ have larger magnitude compared to the normalized distributions $p(x_i), q(x_i) \in [0, 1]$, the $\hat{p}(x_i)$ preserves the long-context ability of target LLM to verify candidates effectively. We empirically validate the robustness of this distribution in §4.5.

For inference, we replace $p(x_i)$ with $\hat{p}(x_i)$ in the acceptance criterion (Eq. (5)). Let $p(x_i) = [\text{w}_j]_{j=1}^{|V|}$ and $\hat{p}(x_i) = [\hat{\text{w}}_j]_{j=1}^{|V|}$ denote the probability vectors over vocabulary $V$. Following Li et al. (2023), we maintain

$$\hat{\text{w}}_k = \text{w}_k, \quad \forall k \in \{v \in [|V|] : \hat{\text{w}}_v < 0.1 \cdot \max_{j \in [|V|]} \hat{\text{w}}_j\}, \quad (9)$$

to prevent distortion in the tail of the distribution.

When rejection occurs, we sample from an adjusted residual distribution:

$$x_i \sim \text{norm}(\max(p(x_i) - \hat{p}(x_i), p(x_i) - q(x_i))). \quad (10)$$

This sampling strategy maintains theoretical guarantees, where we prove in Appx. §B that the resulting tokens follow the same distribution as direct sampling from the original target model $p(x_i)$.

## 3. Experimental Setup

### 3.1. Implementation Details

**Target and Draft LLMs.** RAPID is evaluated across different model scales using LLaMA-3.1 (8B, 70B) and

*Table 1.* Comprehensive evaluation of RAPID against baseline methods across different target-draft model configurations. We report performance on ∞Bench and LongBench v2, along with prefill time and throughput speedup on LongBench v2 (Long, CoT) subset. LC and RAG denote evaluating the target model on long and retrieval contexts, respectively. For RAPID, we evaluate both self-speculation (using same-size RAG drafter) and upward-speculation (using larger RAG drafter) settings. Green/red highlighting indicates better/worse performance compared to LC baseline. **Bold** and underline indicate best and second best metric score.

| Target Model | Method | Draft Model | ∞Bench | | | | LongBench v2 | | Efficiency | |
|---|---|---|---|---|---|---|---|---|---|---|
| | | | En. QA | En. MC | En. Sum | AVG. | Overall | Overall (CoT) | Prefill Time (s) | Speedup |
| LLaMA-3.1-8B | LC | - | 34.58 | 53.28 | 30.14 | 39.33 | 28.0 | 30.4 | 25.89 | 1.00× |
| | RAG | - | 31.91 | 62.01 | 27.27 | 40.40 | 29.2 | 33.4 | **0.36** | **3.35×** |
| | SD | - | 32.90 | 55.90 | 30.11 | 39.64 | 29.4 | 31.0 | 26.37 | 1.63× |
| | MagicDec | - | 29.83 | 52.03 | 30.18 | 37.35 | 29.2 | 30.6 | 26.05 | 0.71× |
| | RAPID | LLaMA-3.1-8B (RAG) | 34.90 | 63.32 | **30.27** | 42.83 | 32.4 | 34.2 | 26.37 | 2.10× |
| | RAPID | LLaMA-3.1-70B (RAG) | **40.94** | **79.04** | 29.96 | **49.98** | **38.8** | **40.2** | 28.04 | 1.14× |
| LLaMA-3.1-70B | LC | - | 36.48 | 68.56 | **30.18** | 45.07 | 31.6 | 36.2 | 160.54 | 1.00× |
| | RAG | - | 38.66 | 76.86 | 27.17 | 47.56 | 38.0 | 39.4 | **2.81** | **4.44×** |
| | RAPID | LLaMA-3.1-70B (RAG) | **40.56** | **81.66** | 29.64 | **50.62** | **40.2** | **40.2** | 163.43 | 2.69× |
| Qwen2.5-7B | LC | - | 16.93 | 66.81 | 30.62 | 38.12 | 30.2 | 33.2 | 20.32 | 1.00× |
| | RAG | - | 20.28 | 75.11 | 25.60 | 40.33 | 31.2 | 33.8 | **0.34** | **6.47×** |
| | RAPID | Qwen2.5-7B (RAG) | 19.81 | 75.98 | 31.64 | 42.48 | 32.0 | 35.4 | 21.62 | 2.65× |
| | RAPID | Qwen2.5-72B (RAG) | **30.10** | **83.84** | **32.21** | **48.72** | **35.6** | **41.2** | 23.45 | 0.93× |
| Qwen2.5-72B | LC | - | 39.21 | 81.66 | 32.45 | 51.11 | 40.0 | 43.9 | 162.42 | 1.00× |
| | RAG | - | 30.72 | 80.22 | 28.63 | 46.52 | 38.8 | 39.8 | **3.09** | **3.60×** |
| | RAPID | Qwen2.5-72B (RAG) | **40.52** | **85.59** | **32.94** | **53.02** | **42.9** | **44.1** | 164.80 | 1.98× |

Qwen2.5 (7B, 72B) as target LLMs. We implement two speculation settings: (1) *self-speculation*, where the RAG drafter matches the target LLM's scale, and (2) *upward-speculation*, where a larger RAG drafter assists a smaller target LLM. For smaller models (LLaMA-3.1-8B, Qwen2.5-7B), we evaluate both settings, while larger models (LLaMA-3.1-70B, Qwen2.5-72B) use self-speculation only. The RAG drafter generates $\gamma = 10$ tokens per step for target LLM verification. We search $\eta$ in Eq. (6) between $\{5, 10, 20\}$ for self-speculation and $\{40, 50\}$ for upward-speculation, which would be further investigated in §4.5.

**RAG Setup.** The long context is segmented into 512-token chunks and embedded using BGE-M3 (Chen et al., 2024b). We retrieve top-$k$ segments based on cosine similarity with the query embedding, filtering out segments below a 0.3 similarity threshold. The retrieval context length is bounded between 4096 tokens and 1/24 of the input length.

### 3.2. Evaluation Protocol

**Baselines.** We compare our RAPID with baselines including: (1) long-context target LLM (LC), where the target LLM in RAPID directly generates responses upon long context; (2) RAG, where the target LLM generates responses upon retrieval context of draft LLM input in RAPID; (3) naive Speculative Decoding (SD), which involves identi-

cal target and draft LLMs with RAPID but using the naive long-context target distribution; (4) MagicDec (Chen et al., 2024a), which utilizes the StreamingLLM (Xiao et al., 2023) to compress the KV cache of draft model. We set the KV cache size as 4096 and sink tokens as 4.

**Benchmarks.** We evaluate our RAPID with baselines on two benchmarks: (1) ∞**Bench**. We evaluate our method with baselines on three realistic tasks in this benchmark: long-book question answering (En.QA, metric: F1), multi-choice question-answering (En.MC, metric: accuracy), and summarization (En.Sum, metric: ROUGE-L-Sum). The context length in these tasks are beyond 100K. (2) **LongBench v2**, which involves multi-choice tasks across various context lengths from 8K to 2M words. We apply middle truncation following benchmark setup to ensure the context length within 128K tokens.

**Evaluation Setup** We conduct efficiency evaluations using the LongBench v2 (Long, CoT) subset, where each example involves 120K (tokens) context length after truncation and 1K maximum generation tokens. Efficiency metrics include: (1) *prefill time* and (2) *speedup*, computed as the ratio of method decoding throughput to LC throughput, both averaged across the subset. Additional experimental details are provided in Appx. §C.

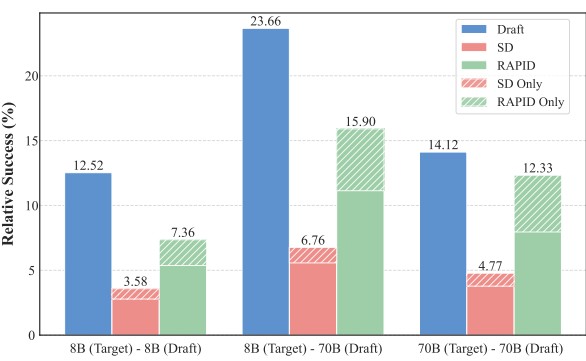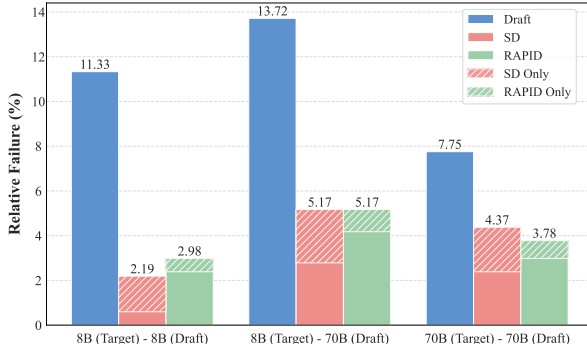

*Figure 2.* Relative performance to target LLMs across different target-draft model configurations of LLaMA-3.1 series on LongBench v2 (Overall). RAPID integrates both benefits from target and draft LLMs, hence achieving higher relative success rate (benefits from draft) without increasing failure rate (benefits from target). Relative success represents correct predictions made by each method but missed by the target LLM. Relative failure represents correct predictions by the target LLM but missed by each method. "SD Only" and "RAPID Only" indicate correct (or wrong) predictions made exclusively by SD and RAPID where both target and draft models cannot attain.

## 4. Results and Analyses

### 4.1. Main Results

We evaluate RAPID against baselines across different model scales and benchmarks. The results in Table 1 demonstrate the effectiveness of RAPID in both improving generation quality and efficiency for long-context inference.

**RAPID integrates benefits from both target LLM and RAG drafter through self-speculation.** In the self-speculation setting, where RAPID uses same-scale models for target and draft, consistent improvements are observed across model families. For LLaMA-3.1-8B, RAPID with self-speculation achieves superior performance on ∞Bench (42.83 vs 39.33 LC, 40.40 RAG) and LongBench v2 (34.2% vs 30.4% LC, 33.4% RAG). Similar improvements are seen for LLaMA-3.1-70B (50.62 vs 45.07 LC, 47.56 RAG on ∞Bench) and Qwen2.5 series. Notably, RAPID effectively integrates the complementary strengths of LC and RAG approaches - while RAG shows superior performance on certain tasks (e.g., En.MC: 79.04% vs 53.28% LC for LLaMA-3.1-8B), LC demonstrates advantages in others (e.g., En.QA: 34.58% vs 31.91% RAG). RAPID successfully captures these complementary benefits during inference, consistently achieving better or comparable performance to the stronger of its two components. Compared to existing speculative decoding approaches including naive SD and MagicDec, RAPID demonstrates superior performance through this effective integration mechanism.

**Larger RAG drafters further boost performance through effective knowledge transfer.** Beyond self-speculation, RAPID enables a unique upward-speculation mechanism where larger models serve as RAG drafters while maintaining efficiency. This setting yields even more substantial improvements: LLaMA-3.1-8B with 70B RAG

drafter achieves 49.98 on ∞Bench and 40.2% overall accuracy on LongBench v2, surpassing not only its self-speculation results but even the LC performance of LLaMA-3.1-70B (36.2%). Similar patterns emerge for Qwen2.5-7B with 72B RAG drafter, where the performance gains (48.72 vs 42.48 on ∞Bench) demonstrate the effectiveness of RAPID in leveraging and integrating knowledge from larger models through the retrieval-augmented speculation.

**RAPID demonstrates $> 2\times$ speedup for long-context inference.** In self-speculation settings, RAPID achieves significant speedup over LC baseline (2.10× for LLaMA-3.1-8B, 2.69× for LLaMA-3.1-70B), and significantly surpasses naive SD and MagicDec. When employing upward-speculation with larger drafters, RAPID still maintains comparable throughput [1] (1.14× for LLaMA-3.1-8B with 70B drafter, 0.93× for Qwen2.5-7B with 72B drafter) while substantially improving generation quality. While pure RAG shows highest throughput (e.g., 3.35× speedup for LLaMA-3.1-8B), its performance can be significantly compromised in certain scenarios (e.g., En.QA accuracy drops from 39.21 to 30.72 for Qwen2.5-72B). In contrast, RAPID effectively maintains competitive throughput while consistently achieving superior generation quality across different settings.

### 4.2. Benefits Integration Analysis

**RAPID incorporates benefits from RAG drafter while maintaining target model capabilities.** To analyze how RAPID integrates the strengths of both RAG drafter and target LLM, we examine the relative success and failure of RAG drafter, SD, and RAPID on LongBench v2. As shown in Figure 2, RAPID successfully handles additional cases where the target LLM fails by incorporating beneficial

---

[1] Note that upward-speculation requires extra GPUs to serve the RAG drafter like regular SD.

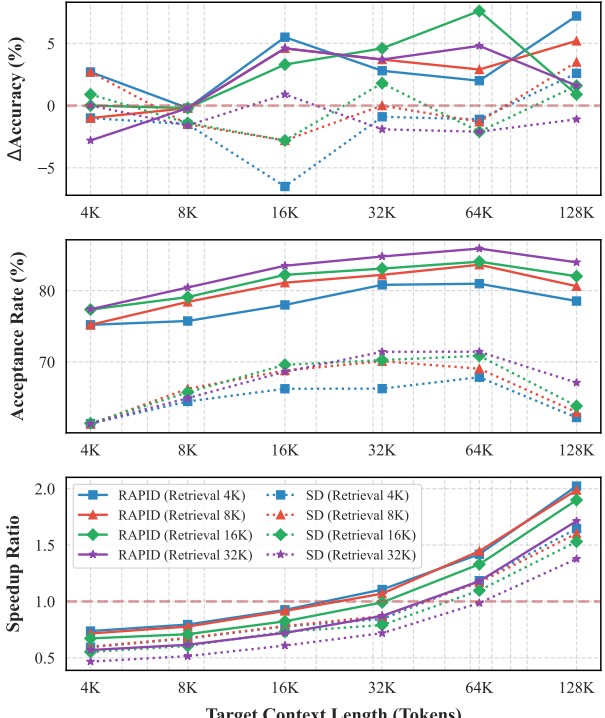

*Figure 3.* Impact of context and retrieval lengths on RAPID (self-sepculation) performance and efficiency based on LLaMA-3.1-8B. RAPID consistently outperforms naive SD and achieves speedup beyond 32K context length with moderate retrieval lengths ($\leq$16K).**Top:**$\Delta$Accuracy indicates the accuracy margins on Long-Bench v2 (Long, CoT) subset over target LLM. **Middle:** Acceptance rate indicating the proportion of accepted draft tokens. **Bottom:** Speedup ratio compared to target LLM inference ($>$ 1 indicates acceleration).

knowledge from the RAG drafter. Meanwhile, RAPID maintains the capabilities of target LLM, exhibiting significantly lower failure rates compared to using RAG drafter alone. This combination of gains from RAG drafter with minimal degradation of target LLM capabilities enables RAPID to outperform both target and draft models. Furthermore, the gains from RAG drafter in RAPID substantially exceed those in naive SD, demonstrating the effectiveness of our retrieval-augmented target distribution in Eq. (6).

**RAPID exhibits capabilities beyond individual target/draft LLMs.** Most notably, we observe an "emergent phenomenon" where RAPID successfully handles cases that both the target LLM and RAG drafter fail individually (shown as "RAPID Only" in Figure 2). Specifically, this emergent accuracy mass grows more pronounced as RAG drafters become stronger, from LLaMA-3.1-8B to LLaMA-3.1-70B. This suggests that RAPID not only combines the strengths of both models but also enables new capabilities through their synergistic interaction. The phenomenon becomes particularly evident in the upward-speculation setting,

*Table 2.* Evaluation on multi-turn dialogue generation with extended chat history for LLaMA-3.1-8B as both target and draft LLM. Quality scores (1-10) are rated by GPT-4-Turbo-1106 using LLM-as-a-Judge protocol.

| | Quality | Acceptance Rate (%) | Throughput |
|---|---|---|---|
| Target LLM | 2.82 | - | $10.64_{\pm0.98}$ |
| RAG Drafter | 3.95 | - | $\mathbf{40.49_{\pm0.47}}$ |
| SD | 2.94 | $56.34_{\pm0.13}$ | $14.07_{\pm3.08}$ |
| RAPID | **4.21** | $\mathbf{76.94_{\pm0.13}}$ | $18.18_{\pm3.23}$ |

*Table 3.* Robustness study of RAPID with different draft influence parameter $\eta$. Results show performance gains ($\Delta$Accuracy) and speedup ratios on LongBench v2 (Long, CoT) subset using LLaMA-3.1-8B as target LLM, with LLaMA-3.1-8B and LLaMA-3.1-70B as RAG drafters under unrelated retrieval context.

| $\eta$ | LLaMA-3.1-8B (Draft) | | LLaMA-3.1-70B (Draft) | |
|---|---|---|---|---|
| | $\Delta$Accuracy | Speedup | $\Delta$Accuracy | Speedup |
| 0 | 1.20 | 1.62$\times$ | -1.30 | 0.67$\times$ |
| 5 | 2.80 | 1.75$\times$ | 0.40 | 0.69$\times$ |
| 10 | 1.60 | 1.77$\times$ | 1.20 | 0.72$\times$ |
| 20 | 1.20 | 1.78$\times$ | 4.40 | 0.75$\times$ |
| 30 | -2.40 | 2.07$\times$ | 6.60 | 0.80$\times$ |
| 40 | -2.60 | 2.08$\times$ | 6.60 | 0.84$\times$ |
| 50 | -6.30 | 2.10$\times$ | 6.00 | 0.87$\times$ |

where the stronger RAG drafter facilitates more sophisticated knowledge transfer during inference.

### 4.3. Impact of Context and Retrieval Length.

**RAPID demonstrates effectiveness across various context configurations.** We analyze how RAPID performs under varying target context lengths and RAG drafter retrieval lengths, as shown in Figure 3. The results demonstrate consistent advantages of RAPID over naive SD across all configurations. First, RAPID achieves significantly better performance gains (2-8% $\Delta$Accuracy) over the long-context baseline compared to the marginal or negative gains (-5-2%) of naive SD. This superior performance is accompanied by consistently higher acceptance rates (75-85% versus 60-70%) and better speedup ratios across all context and retrieval lengths configurations.

**RAPID achieves speedup for long-context inference beyond 32K.** The impact of retrieval length reveals an interesting efficiency-effectiveness trade-off. In terms of computational efficiency, RAPID achieves acceleration (speedup $>$ 1.0$\times$) when the target context length exceeds 32K, while SD requires contexts beyond 64K to demonstrate speedup. For retrieval length, while longer retrieval contexts generally lead to higher acceptance rates (up to 85%), the speedup ratio is not necessarily increasing. Specifically, retrieval lengths of 4K and 8K achieve nearly identical speedup ratios, indicating minimal overhead in this scope. However,

when retrieval length exceeds 16K, the increased computational overhead from longer draft contexts becomes apparent and impacts the overall speedup. These findings suggest that RAPID achieves remarkable efficiency when accelerating long-context inference beyond 32K tokens upon moderate retrieval length within 16K.

## 4.4. Generation Quality Analysis

**RAPID achieves superior generation quality and throughput in real-world application.** To evaluate the effectiveness of RAPID in practical long-context applications, we assess its performance on multi-turn dialogue generation. We construct a challenging evaluation dataset by adapting MT-Bench-101 (Bai et al., 2024a): for each of the first 100 samples, we preserve their last-turn queries while distributing their previous conversation context within a longer chat history comprising additional dialogue turns from another 500 samples in MT-Bench-101. The resulting chat history is of around 122K tokens length. This setup tests the ability of models to maintain coherence and relevance while processing extensive dialogue history.

As shown in Table 2, RAPID demonstrates substantial improvements across all metrics. Using GPT-4-Turbo-1106 as evaluator following LLM-as-a-Judge (Zheng et al., 2023), RAPID achieves a generation quality score of 4.21, significantly outperforming the target LLM (2.82), RAG drafter (3.95) and naive SD (2.94). This quality improvement comes with a robust acceptance rate of 76.94% (vs. 56.34% for SD) and enhanced throughput of 18.18 tokens/second ($1.7\times$ speedup over target LLM), demonstrating practical advantages of RAPID in real-world long-context applications.

## 4.5. Robustness to Retrieval Quality

**RAPID shows robustness to retrieval quality, which is further enhanced by stronger drafter.** To assess the robustness of RAPID regarding retrieval quality, we conduct stress tests by deliberately using unrelated retrieval context (using the context of first sample from LongBench v2 for all samples) while varying the knowledge transfer parameter $\eta$ in Eq. (6). As shown in Table 3, with self-speculation (LLaMA-3.1-8B drafter), RAPID maintains performance gains ($\Delta$Accuracy $> 0$) and improved efficiency (speedup $1.62\times$-$1.78\times$) when $\eta \leq 20$, even with irrelevant retrieval context. However, when $\eta > 20$, the RAG drafter may overly impact the target distribution, leading to performance degradation. Moreover, upward-speculation with LLaMA-3.1-70B as drafter demonstrates even better robustness, maintaining positive performance gains (up to 6.60%) across all $\eta$ values despite totally unrelated retrieval context. This increased resilience suggests that RAPID effectively leverages the inherent capabilities of stronger RAG drafters, maintaining reliable performance even under suboptimal retrieval quality.

## 5. Related Work

**Speculative Decoding** Speculative Decoding (Chen et al., 2023; Leviathan et al., 2023) accelerates LLM inference by leveraging smaller draft models to propose multiple tokens for single-pass validation. REST (He et al., 2024b) extends the drafting mechanism by retrieving possible continuation from a built corpus rather than generating with a draft LLM. Ouroboros (Zhao et al., 2024) proposes producing longer and more acceptable candidates from draft LLM per step based on draft phrases. Inspired by the speculation mechanism, Speculative RAG (Wang et al., 2024) proposes a parallel draft-then-verify mechanism to improve RAG quality. Recent works like TriForce (Sun et al., 2024) and MagicDec (Chen et al., 2024a) attempt to extend SD to long-context scenarios through KV cache compression techniques (Xiao et al., 2023). However, such compression approaches often result in weakened draft models with limited speedup in complex applications. In contrast, RAPID adopts RAG drafters that maintain both high-quality speculation and substantial speedup in various applications.

**Long-Context Inference Speedup** Research on accelerating long-context inference has primarily focused on two directions: optimizing KV cache operations through selective retention (Xiao et al., 2023; Kang et al., 2024; Zhang et al., 2023) or quantization (Sheng et al., 2023; Liu et al., 2024b; He et al., 2024a), and exploring prompt compression methods (Chevalier et al., 2023; Jiang et al., 2023; Pan et al., 2024). While these approaches improve efficiency, they often compromise contextual information without quality guarantees (Zhang et al., 2024). RAPID addresses this limitation by leveraging SD to maintain generation quality through explicit verification from long-context LLMs, providing a more reliable balance between efficiency and performance.

**RAG and Long-Context LLMs** Recent studies have revealed complementary strengths between RAG and long-context LLMs, with substantial prediction overlap despite different performance characteristics (Li et al., 2024b;a). While long-context LLMs excel in document-based tasks, RAG shows advantages in scenarios like dialogue-based question-answering. Previous attempts to combine these approaches, such as self-reflection routing (Li et al., 2024b) and step-by-step RAG enhancement (Yue et al., 2024), rely heavily on task-specific prompt engineering. RAPID provides a more principled solution by directly integrating RAG benefits into the decoding process, enabling dynamic adaptation while preserving advantages of both paradigms.

## 6. Conclusion

In this work, we introduce RAPID, a novel decoding method that bridges the efficiency gap of speculative decoding (SD) in long-context inference while enhancing generation quality through retrieval-augmented speculation. The key of RAPID lies in leveraging RAG drafters to enable efficient speculation for long-context target LLMs, along with a retrieval-augmented target distribution that effectively integrates knowledge from potentially stronger drafters. Through extensive experiments, we demonstrate that RAPID successfully achieves both computational efficiency and improved generation quality across different model scales and tasks. Specifically, RAPID enables more than $2\times$ speedup while maintaining performance advantages in self-speculation settings, and achieves substantial quality improvements through upward-speculation with stronger RAG drafters. These results establish RAPID as a practical solution for accelerating long-context inference with improved generation quality.

## Acknowledgments

This project was partially supported by the Singapore Ministry of Education Academic Research Fund Tier 1 (Award Number: T1 251RES2514) and DAMO Academy Research Intern Program.

## Impact Statement

This paper presents work whose goal is to advance the field of Machine Learning. There are many potential societal consequences of our work, none which we feel must be specifically highlighted here.

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

# A. Proof of Theorem 1

We analyze the gradient of the knowledge distillation loss with respect to the target model's logits. The distillation loss with temperature $\mathcal{T}$ is defined as:

$$\begin{aligned}
\mathcal{L} &= \mathcal{T}^2 \cdot \mathrm{KL}(q(x)||p(x)) \\
&= \mathcal{T}^2 \sum_j q(x_j) \log \frac{q(x_j)}{p(x_j)}
\end{aligned} \tag{11}$$

where the target distribution $p(x)$ is parameterized by logits $z$ through softmax:

$$p(x_j) = \frac{\exp(z_j/\mathcal{T})}{\sum_k \exp(z_k/\mathcal{T})} \tag{12}$$

**Theorem:** The gradient of the distillation loss with respect to logit $z_i$ is:

$$\frac{\partial \mathcal{L}}{\partial z_i} = -\mathcal{T}[q(x_i) - p(x_i)] \tag{13}$$

**Proof:** We derive this gradient through the following steps:

1) First, expand the derivative using the chain rule:

$$\frac{\partial \mathcal{L}}{\partial z_i} = \mathcal{T}^2 \sum_j q(x_j) \frac{\partial}{\partial z_i}[\log q(x_j) - \log p(x_j)] \tag{14}$$

2) Note that $q(x_j)$ is independent of $z_i$:

$$= -\mathcal{T}^2 \sum_j q(x_j) \frac{\partial}{\partial z_i} \log p(x_j) \tag{15}$$

3) Expand the log probability:

$$= -\mathcal{T}^2 \sum_j q(x_j) \frac{\partial}{\partial z_i} \left[ \frac{z_j}{\mathcal{T}} - \log \sum_k \exp(z_k/\mathcal{T}) \right] \tag{16}$$

4) Apply the derivative using the Kronecker delta $\delta_{ij}$:

$$= -\mathcal{T}^2 \sum_j q(x_j) \left[ \frac{\delta_{ij}}{\mathcal{T}} - \frac{1}{\mathcal{T}} \frac{\exp(z_i/\mathcal{T})}{\sum_k \exp(z_k/\mathcal{T})} \right] \tag{17}$$

5) Simplify using the definition of $p(x_i)$:

$$= -\mathcal{T} \sum_j q(x_j)[\delta_{ij} - p(x_i)] \tag{18}$$

6) The sum over $j$ with $\delta_{ij}$ selects only $q(x_i)$:

$$= -\mathcal{T}[q(x_i) - \sum_j q(x_j)p(x_i)] \tag{19}$$

7) Since $\sum_j q(x_j) = 1$, we obtain our final result:

$$= -\mathcal{T}[q(x_i) - p(x_i)] \tag{20}$$

This gradient shows that the distillation loss pushes the target distribution $p(x)$ towards the draft distribution $q(x)$ with strength proportional to the temperature $\mathcal{T}$. $\square$

## B. Correctness of RAPID's Residual Distribution

We prove that for RAPID's retrieval-augmented speculative decoding, when rejection occurs, sampling from the distribution

$$x_i \sim \texttt{norm}(\max(p(x_i) - \hat{p}(x_i), p(x_i) - q(x_i))) \tag{21}$$

maintains the target distribution $p(x_i)$, where:

$$p(x_i) = p_{\boldsymbol{\phi}}(x_i | [\mathcal{C}; x_{<i}]) \text{ (target distribution)} \tag{22}$$

$$q(x_i) = q_{\boldsymbol{\psi}}(x_i | [\mathcal{C}^{\text{S}}; x_{<i}]) \text{ (RAG drafter distribution)} \tag{23}$$

$$\hat{p}(x_i) = \text{softmax}(\hat{z}(x_i)/T) \text{ (retrieval-augmented target)} \tag{24}$$

**Proof:** Let $x'$ be a candidate token. Under RAPID's rejection sampling scheme:

1) For a token $x'$ proposed by the draft model, the acceptance criterion is:

$$r \leq \min(1, \frac{\hat{p}(x')}{q(x')}) \tag{25}$$

where $r \sim U(0, 1)$

2) This leads to an acceptance probability:

$$P(\text{accept}|x') = \min(q(x'), \hat{p}(x')) \tag{26}$$

3) The residual probability mass that needs to be redistributed upon rejection is:

$$p(x') - \min(q(x'), \hat{p}(x')) = \max(p(x') - q(x'), p(x') - \hat{p}(x')) \tag{27}$$

4) Let $\beta$ be the total acceptance probability:

$$\beta = \sum_{x'} \min(q(x'), \hat{p}(x')) \tag{28}$$

5) Therefore, upon rejection, we must sample from:

$$p'(x') = \frac{p(x') - \min(q(x'), \hat{p}(x'))}{\sum_{x'}(p(x') - \min(q(x'), \hat{p}(x')))} = \frac{p(x') - \min(q(x'), \hat{p}(x'))}{1 - \beta} \tag{29}$$

This residual distribution ensures that for any token $x'$:

$$P(x = x') = \min(q(x'), \hat{p}(x')) + (1 - \beta)\frac{p(x') - \min(q(x'), \hat{p}(x'))}{1 - \beta} = p(x') \tag{30}$$

## C. Evaluation Setup

We conduct comprehensive evaluations across different model scales and configurations. We use temperature values of 1.0 and 0.1 for $\infty$Bench and LongBench v2, respectively. For base-scale models (LLaMA-3.1-8B and Qwen2.5-7B), we evaluate RAPID's self-speculation capabilities against multiple baselines including naive Speculative Decoding, MagicDec, Long Context (LC), and RAG implementations, using a single NVIDIA A800 80GB GPU.

For large-scale models (LLaMA-3.1-70B and Qwen2.5-72B), self-speculation experiments are conducted using a distributed setup with 8×A800 80GB GPUs. In upward-speculation settings, we employ a hybrid configuration where the target models (LLaMA-3.1-8B/Qwen2.5-7B) operate on a single A800 80GB GPU, while leveraging an additional 7×A800 80GB GPUs to accommodate the larger RAG drafter.

## D. More Efficiency Analyses

### D.1. FLOPs Comparison

We present a detailed comparison of floating-point operations (FLOPs) per generation step (producing $\gamma$ tokens) in Table 4, analyzing our RAPID method against baseline approaches. Let $T$ denote the number of parameters in the target model and $L$ represent the long context length. For the draft model, we define:

- $D$: Number of parameters

- $L^R$: Retrieval length for draft LLM input

The key parameters for speculative generation include:

- $\gamma$: Number of tokens generated by the draft model per step

- $\beta^{SD}$: Expected acceptance rate for standard speculative decoding

- $\beta^{RAPID}$: Expected acceptance rate for RAPID

Our analysis reveals that while all methods scale linearly with the target model size $T$, RAPID achieves superior efficiency through its higher acceptance rate ($\beta^{RAPID} > \beta^{SD}$), which directly reduces the amortized FLOPs per generated token.

Table 4. FLOPs comparison for different methods per step.

| Method | FLOPs |
|---|---|
| Long Context | $2\gamma TL + \gamma^2 T$ |
| RAG Drafter | $2\gamma DL^R + \gamma^2 D$ |
| SD | $\frac{2\gamma DL^R + \gamma^2 D + 2T(L+\gamma)}{\beta^{SD}}$ |
| RAPID | $\frac{2\gamma DL^R + \gamma^2 D + 2T(L+\gamma)}{\beta^{RAPID}}$ |

### D.2. Overhead of RAG

Unlike regular RAG pipeline, which builds indexes for a large external corpus (hundreds of millions of documents), we only index/retrieve the chunks for the input long context (<128K) on-the-fly during inference. Therefore, the RAG component latency in our method will become marginal compared to the inference latency over long context. Table 5 presents the average latency (in seconds) for each component of RAPID on LongBench v2 (Long, CoT) using LLaMA-3.1-8B and LLaMA-3.1-70B in self-speculative mode.

Table 5. Latency of RAPID Components on LongBench v2 (Long, CoT)

| Model | RAG Pipeline (s) | Prefill (s) | Generation (s) |
|---|---|---|---|
| LLaMA-3.1-8B-RAPID | 1.43 | 26.37 | 32.25 |
| LLaMA-3.1-70B-RAPID | 1.43 | 163.43 | 121.76 |

## E. More Results

### E.1. Comparison with TriForce

TriForce was not included in Table 1 since it is not directly compatible with modern LLMs using Grouped Query Attention (GQA) (Ainslie et al., 2023a). We conducted comparisons on LWM-Text-Chat-128K (Liu et al., 2024a) (based on LLaMA2-7B (Touvron et al., 2023)), with a retrieval budget of 4096 tokens, a chunk size of 8, and a draft cache budget of 256 for TriForce. Table 6 shows the performance and speedup of the decoding in LongBench v2 (Long, CoT).

*Table 6.* Comparison of RAPID and TriForce on LWM-Text-Chat-128K in LongBench v2 (Long, CoT) task.

| Model | Accuracy | Speedup |
|---|---|---|
| LWM-Text-Chat-128K | 18.4 | 1.00 |
| TriForce | 18.0 | 1.27 |
| RAPID | 21.6 | 2.56 |

While TriForce achieves modest efficiency gains, RAPID delivers superior speedup and performance. TriForce relies on chunk-wise attention scores for information recall, but high attention scores do not always correlate with semantic relevance, e.g., initial tokens may act as "attention sinks" despite lacking meaningful content (Xiao et al., 2023). In contrast, our RAPID drafter prioritizes semantically relevant information, resulting in a higher acceptance rate and greater speedup for complex tasks.

### E.2. Comparison with MInference

We evaluated MInference (Jiang et al., 2024) against our RAPID using LLaMA-3.1-8B on the LongBench v2 (Long, CoT) task. Table 7 reports the performance, prefill time (in seconds), and decoding speedup relative to the LLaMA-3.1-8B.

*Table 7.* Comparison of RAPID and MInference on LLaMA-3.1-8B in LongBench v2 (Long, CoT) task.

| Model | Accuracy | Prefill Time (s) | Speedup |
|---|---|---|---|
| LLaMA-3.1-8B (Baseline) | 30.4 | 25.89 | 1.00 |
| MInference | 30.9 | 9.10 | 0.62 |
| RAPID | 34.2 | 26.37 | 2.10 |

MInference significantly reduces prefill time, showcasing its efficiency in the initial processing phase. However, RAPID outperforms MInference in overall performance and decoding throughput, achieving a higher speedup. We note that sparse attention, as utilized by MInference, is orthogonal to our approach, suggesting that integrating sparse attention with RAPID could further enhance efficiency.

