# OpenReview forum: "RAPID: Long-Context Inference with Retrieval-Augmented Speculative Decoding"
_ICML.cc/2025/Conference — ICML 2025 spotlightposter_

### Official Review · Reviewer_KbZ7 · 2025-03-12

**Overall Recommendation:** 4

**Summary:**

This work proposes RAPID, a variation on the typical speculative decoding framework for long-context tasks by using relatively large draft models that use RAG to compress the context. The quality and performance of RAPID is further boosted by using a “retrieval-augmented target distribution” which modifies the target model’s output distribution by multiplying the original target model logits by the difference of the draft and target logit distributions; the influence of the draft model logits is controlled with a hyperparameter, $\eta$. The benefits of RAPID for long-contexts are demonstrated with models from two model families and two long-context benchmarks. RAPID achieves the highest accuracy and overall throughput increases compared to the other baselines examined.

**Claims And Evidence:**

* RAPID claims to outperform both naive SD and MagicDec. However, only Llama-8B is used to establish this relationship. It would be more convincing to include the other models examined.
* The authors claim that TriForce results in “weakend draft models”. As TriForce is not used as a baseline in the paper, this is an unsubstantiated claim. Notably, TriForce’s second tier retrieval cache based drafter is similar to RAPID’s use of RAG for drafting.
* Otherwise, the claims made appear to be well founded.

**Essential References Not Discussed:**

In my opinion [2] should be included as a seminal work that inspired much of the following speculative decoding literature.

[2] M. Stern, N. Shazeer, and J. Uszkoreit, “Blockwise Parallel Decoding for Deep Autoregressive Models,” in Advances in Neural Information Processing Systems, Curran Associates, Inc., 2018.

**Experimental Designs Or Analyses:**

* My primary concern with the experimental design and analyses is the lack of baseline comparisons across all models.
* Further, additional baseline models should be considered such as specialized long-context models such as Qwen-1M and alternative approaches to accelerating long-context generation such as sparse attention (MInference or similar). While I believe the experimental results are still valuable without these direct comparisons, more robust baselines would better convince me that RAPID offers advantages over these competing methods.
* The ablation studies are somewhat limited. While the authors analyze the impact of context and retrieval length and the effect of retrieval length, I think additional ablations would be helpful. For example, it would be useful to see the impact of different retrieval strategies or different choices of the draft model or how sensitive the cosine similarity threshold is to the overall methods performance and quality.
* The generation quality experiment relies on LLM-as-a-Judge evaluation using GPT-4 Turbo. This evaluation method is subjective and can be unreliable. An additional judge or human-verification would improve reliability of these results.
* The generation quality analysis also relies on a synthetic dataset in which unrelated dialogs are inserted into a multi-turn chat context. This seems like a setting in which RAG would be disproportionality well suited to given that the unrelated dialogs would have low similarity to the context of interest. This is distinct from real-world long contexts which typically contain similar themes / topics throughout.

**Methods And Evaluation Criteria:**

Generally, the methods and evaluation criteria are sensible for the method proposed.

**Other Comments Or Suggestions:**

* L065: I believe this cross reference should be for Figure 3 not 1?
* L074: Self speculation is not a novel contribution of this work and should be removed from the claim that the work introduces it as a “new paradigm”.
* L086: I believe this should be DRAM not SRAM. The memory i/o latency bottleneck is typically from DDR memory to the streaming multiprocessor rather than from shared memory.
* Table 1 first row of Qwen-72B results should have no shading, appears light pink.
* L328: “Infernce”

**Other Strengths And Weaknesses:**

## Strengths:
* Important and timely topic
* Strong empirical results, particularly with upward-speculation in which the Llama-8B target + 70B drafter outperforms the 70B LC model both in terms of quality and performance.
* The proposed retrieval-augmented target distribution is novel and appears to address one of the main pitfalls of speculative decoding in that the target model is assumed to represent the ground truth for the drafter.
* The paper provides a good combination of algorithmic development, theoretical analysis, and experimental evaluation.

## Weaknesses:
* RAPIDs has several hyperparameters: the number of draft tokens per round, $\eta$, the cosine similarity threshold for retrieval, the retrieval length, and the compression ratio. It’s not clear how each of these values were selected nor how sensitive the method is to each of these. Only $\eta$ is studied in detail and in a somewhat unrelated setting to the main results since an unrelated context is used.
* Additional baselines should be considered to compare RAPIDs to other speculative decoding methods like TriForce and other approaches in the long-context literature such as sparse attention.

**Questions For Authors:**

1. How does RAPID compare with specialized LC models such as Qwen-1M? How does RAPID compare with TriForce? What are the results for SD and MagicDec on the other models not reported? This is the key question to answer to improve my score.
2. How were the following hyperparameters selected: the number of draft tokens per round, the cosine similarity threshold for retrieval, retrieval length, and the compression ratio. How sensitive is RAPID to changing these parameters?
3. The robustness analysis in Section 4.5 is surprising. I would have expected a larger degradation in quality given that completely unrelated context is used. Could the authors speculate on where the latency / accuracy gains are coming from for these results given that the draft model is preventing from attending the correct context?

**Relation To Broader Scientific Literature:**

* Accelerating long-context generation is a topic that has seen significant interest of late. MagicDec and TriForce are the leading examples which approach the problem from the perspective of speculative decoding. Competing methodologies such as sparse attention for prefill (MInference etc.) and sparse decoding have also seen significant interest.
* The combination of speculative decoding and RAG appear for long-context generation is related to but distinct from other recent approaches that rely on approximated attention using truncated queries (ie.., the last 64 tokens in a long prompt) or compressed / pooled k/v vectors (RetrievalAttention, Quest).
* With respect to prior literature, I believe the most novel contribution is the proposed retrieval-augmented target distribution. Combining RAG with the draft model is closely related to TriForce and MagicDec in which the draft model uses a compressed KV cache of some kind.
* This work shares some similarity with [1] but is motivated specifically for long-context generation and uses a distinctly different approach.

[1] https://arxiv.org/abs/2407.08223

**Theoretical Claims:**

I did not review the proofs in the supplementary materials. The claims regarding the proposed retrieval-augmented target distribution appear to be sensible.

---

> ### Author Rebuttal · Authors · 2025-04-01
>
> Dear Reviewer KbZ7,
>
> We sincerely appreciate your thorough review of our paper. Your constructive feedback will help us strengthen this work. Below are our responses to your concerns:
>
> ---
>
> ## 1. Comparison with LC models like Qwen-1M
>
> We've evaluated RAPID on Qwen2.5-7B-1M (released post-submission). Results show RAPID on Qwen2.5-7B-Instruct is comparable to Qwen2.5-7B-1M (35.4 vs 35.6), while RAPID on Qwen2.5-7B-1M further improves both efficiency and performance:
>
> |  | Overall (CoT) | Speedup |
> | --- | --- | --- |
> | Qwen2.5-7B-1M | 35.6 | 1 |
> | - RAPID ($\eta = 5$) | 38.4 | 2.01 |
>
> ---
>
> ## 2. Comparison with Triforce
>
> Triforce wasn't included because **it can't be directly applied to new LLMs with GQA**. We've now conducted comparisons on LWM-Text-Chat-128K (based on LLaMA2-7B), setting retrieval budget at 4096, chunk size at 8, and draft cache budget at 256 for Triforce:
>
> |  | Overall (CoT) | Speedup |
> | --- | --- | --- |
> | LWM-Text-Chat-128K | 18.4 | 1 |
> | - Triforce | 18.0 | 1.27 |
> | - RAPID | 21.6 | 2.56 |
>
> **While Triforce achieves efficiency gains, RAPID demonstrates more significant speedup and performance improvements**. Triforce recalls information based on chunk-wise attention scores, but higher attention scores don't necessarily indicate greater semantic relevance (e.g., initial tokens often attract high attention as "sinks" despite lacking semantic importance [1]). Our RAD drafter better recalls semantically relevant information, hence achieving higher acceptance rate and speedup for more challenging tasks.
>
> [1] Efficient Streaming Language Models with Attention Sinks
>
> ---
>
> ## 3. Additional Results for MagicDec and SD
>
> We haven’t included results of MagicDec and SD on more models since they achieve quite similar efficiency gains (significantly below our method) across models, while they aren’t designed for performance improvement. We now include more results below.
>
> |  | Overall (CoT) | Speedup |
> | --- | --- | --- |
> | Qwen2.5-7B-SD | 29.2 | 1.83 |
> | -MagicDec | 30.0 | 0.71 |
> | Qwen2.5-70B-SD | 43.7 | 1.59 |
> | -MagicDec | 43.5 | 0.65 |
> | LLaMA3.1-70B-SD | 35.3 | 1.75 |
> | -MagicDec | 34.8 | 0.73 |
>
> ---
>
> ## 4. Comparison with Sparse Attention (MInference)
>
> Per your suggestion, we've compared with MInference on LLaMA-3.1-8B. Results show **MInference achieves impressive prefill speedup, while RAPID demonstrates significant performance and decoding throughput advantages**:
>
> |  | Overall (CoT) | Prefill Time | Speedup |
> | --- | --- | --- | --- |
> | LLaMA-3.1-8B | 30.4 | 25.89 | 1 |
> | -MInference | 30.9 | **9.10** | 0.62 |
> | -RAPID | 34.2 | 26.37 | **2.10** |
>
> We believe sparse attention is orthogonal to our work and combining it with RAPID holds potential for future research.
>
> ---
>
> ## 5. Retrieval Hyperparameter Settings
>
> We haven't extensively tuned hyperparameters except for $\eta$, as our goal is to propose a generally effective method rather than overfitting benchmarks. The RAG hyperparameters only affect retrieval quality, and RAPID has demonstrated such robustness (Section 4.5). To clarify our selection criteria:
>
> - **Cosine similarity threshold (0.3)** was selected from {0.1, 0.2, 0.3, 0.4, 0.5} for RAG on LLaMA-3.1-8B, with overall scores of {28.8, 29.2, 29.2, 29.0, 29.0}.
> - **Compression ratio** of 24 (120K/5K) was chosen as retrieval length beyond 5K showed no significant benefits.
>
> For draft tokens per step, we set 10 without tuning. Our ablation study below shows RAPID maintains stable performance with <15 draft tokens, though excessive tokens may reduce performance gains despite increasing throughput.
>
> | # candidates | Overall (CoT) | Speedup |
> | --- | --- | --- |
> | 5 | 34.0 | 1.95 |
> | 10 | 34.2 | 2.1 |
> | 15 | 34.4 | 2.24 |
> | 20 | 32.8 | 2.51 |
>
> ---
>
> ## 6. Robustness Analysis Intuition
>
> The robustness analysis in Section 4.5 was conducted on LongBench v2 (Long, CoT) subset, which involves generating reasoning paths before providing answers. Given some preliminary chains,  we believe a strong drafter is capable of generating continual CoTs with higher quality to introduce the performance gains. In our LLaMA-3.1-8B target / 70B draft analysis, acceptance rates significantly increased after generating 32 tokens despite irrelevant retrieval context, supporting this intuition.
>
> ---
>
> ## 7. Generation Quality Analysis
>
> Our generation quality analysis (Section 4.4) is a pilot experiment demonstrating RAPID's potential effectiveness in real-world applications. We acknowledge that real-world long-context conversations typically contain similar themes/topics, and hope for robust benchmarks with better evaluation metrics beyond LLM-as-judge in the future.  (Human evaluation is both expensive and logistically complex at scale.)
>
> ---
>
> We hope the responses above can address your concerns and contribute to a reconsideration of review score. We also appreciate your careful review to point out some typos/missed references, which will be fixed in a revised version.
>
> Best,
>
> Authors

---

> > ### Comment · Reviewer_KbZ7 · 2025-04-04
> >
> > I thank the authors for their very detailed rebuttal. I have elected to raise my original rating to 4.

---

> > > ### Author Response · Authors · 2025-04-08
> > >
> > > Dear Reviewer,
> > >
> > > Thank you for taking the time to review our rebuttal and for raising your rating. We appreciate your thoughtful consideration of our work and explanations. Your feedback has been valuable in helping us improve our paper.
> > >
> > > Best regards,
> > >
> > > Authors

---

### Official Review · Reviewer_SMJY · 2025-03-13

**Overall Recommendation:** 4

**Summary:**

To enhance the efficiency and effectiveness in long-context scenarios, the paper proposes a method called Retrieval-Augmented Predictive Decoding (RAPID), which aims to address the decline in efficiency and quality of traditional speculative decoding due to memory limitations in long-context reasoning. RAPID integrates RAG , selectively retrieving compressed context  from long documents to generate candidate tokens. This approach reduces computational overhead while maintaining information relevance. Additionally, RAPID incorporates a knowledge distillation mechanism, transferring the knowledge of the RAG Drafter to the target model to form an enhanced target distribution. This not only increases the acceptance rate of high-quality candidates but also maintains theoretical reliability.

**Claims And Evidence:**

Yes, all claims made in the submission are generally supported by clear and convincing evidence.

**Essential References Not Discussed:**

The author's core contribution is the RAG+SD collaborative method. However, this is not an entirely new idea. This paper is not the first to propose the RAG+SD method, yet the author fails to discuss related work in the Introduction and the Related Work sections.

For example:

- REST: Retrieval-Based Speculative Decoding, published in NAACL 2024.
- TRIFORCE: Lossless Acceleration of Long Sequence Generation with Hierarchical Speculative Decoding, published in COLM 2024.
- Ouroboros: Generating Longer Drafts Phrase by Phrase for Faster Speculative Decoding, published in EMNLP 2024.
- Speculative RAG: Enhancing Retrieval-Augmented Generation through Drafting, published in ICLR 2025.

**Experimental Designs Or Analyses:**

In the main experiment 4.1, the author conducted a detailed analysis of the efficiency of RAPID, demonstrating a 2X performance improvement under long context conditions. However, in the analysis section, there is a lack of discussion on the breakdown of time consumption. In particular, it appears that the chunking of context and the construction of indexes are both carried out during the online process, which also consumes a significant amount of time. It is unclear whether this part is included in the overall time calculation and what proportion it accounts for. This issue has a significant impact on the evaluation of RAPID's efficiency.

**Methods And Evaluation Criteria:**

Yes, the proposed methods and evaluation criteria are well-aligned with the problem of long-context inference acceleration. And further detailed evaluation is discussed in  Appendix C.

**Other Comments Or Suggestions:**

The overall writing is smooth, but the abbreviations should have their first letters capitalized. For example, the full form of RAG is not consistent throughout the text.

**Other Strengths And Weaknesses:**

**Strength:**

- The author's choice of research question is highly valuable, and the approach is somewhat innovative. Numerous studies have demonstrated the respective strengths and weaknesses of RAG and direct LLM responses in long-context scenarios. This paper proposes a method to integrate the two in long-context tasks, balancing efficiency and effectiveness.
- The work extends the research on speculative decoding . In traditional SD studies, a smaller-parameter model is typically used as the Drafter. The author proposes using a model with an equivalent or even larger parameter scale but with a smaller context as the Drafter, and introduces knowledge transfer. To some extent, this integrates the idea of knowledge distillation into SD research.

**Weakness:**

- As mentioned in the section on Essential References Not Discussed, there has already been much work on combining RAG and SD. The author's review of related work in this area is not comprehensive, and there is a lack of targeted comparisons. This makes it difficult to fully assess the novelty and innovation of RAPID.
- From a performance perspective, the pipeline is relatively long, and efficiency is crucial. The overhead of each component (e.g., Long-context RAG, Drafter, Target LLM) needs to be better demonstrated in terms of practicality, which will determine the overall applicability of this paper.
- Potential Bias of the RAG Drafter. The candidates generated by the RAG Drafter may be overly reliant on retrieved passages, excessively pruning contextual information and potentially losing too much detail. This could lead to locally optimal rather than globally optimal generated content. The performance of RAPID is highly dependent on retrieval quality, yet the paper does not sufficiently discuss the limitations of the retrieval module and the long-term impact of such biases on long-context reasoning.

**Questions For Authors:**

1. The distinction, rationality, and overhead of using a larger model as the Drafter compared to traditional Speculative Decoding require additional explanation. Using a larger model to enhance and accelerate a smaller model, especially when the smaller model is responsible for validating the speculative tokens generated by the Drafter, raises several questions. If the RAG Drafter is more powerful, would such validation not have a negative impact? On the other hand, when using a larger Drafter for inference, the inference speed itself is not fast. Is the speed gain actually derived from the smaller context? I hope the author can provide a more detailed description of the motivation behind this approach.


2. Recent studies[1,2] have shown that in long-context scenarios, RAG does not always improve the performance of LLMs, especially with more powerful models. Similar conclusions are also validated in this paper. Can the author provide further analysis, such as case studies, to explain the reasons for the performance gains in RAPID? Is it because RAG retrieval makes the information in long-context scenarios more focused? Conversely, in which scenarios do the bad cases occur?

3. When performing inference knowledge transfer, is it necessary to use models from the same family (e.g., Llama series and Qwen series)? If not, that is, if there is a significant difference in token distribution between the teacher model and the student model, will the same effect be achieved or will the gain be weakened? I hope to see a discussion from the author on the generalizability of this approach.

Reference:

 [1] U-NIAH: Unified RAG and LLM Evaluation for Long Context Needle-In-A-Haystack

 [2] LaRA: Benchmarking Retrieval-Augmented Generation and Long-Context LLMs -- No Silver Bullet for LC or RAG Routing

**Relation To Broader Scientific Literature:**

RAPID builds on prior work in speculative decoding, retrieval-augmented generation, and long-context optimization. Key connections include:
1. SD Limitations: RAPID addresses SD’s inefficiency in long contexts by replacing the draft model with a RAG drafter.
2. RAG Integration: By leveraging RAG’s context compression, RAPID avoids KV cache bottlenecks .
3. Knowledge Transfer: The retrieval-augmented target distribution aligns with distillation techniques , enabling upward-speculation.

**Theoretical Claims:**

Theoretical claims are supported by proofs in the appendix.
1. In Appendix A, the gradient derivation for knowledge distillation loss is correct, showing the logits shift from RAG drafter to target model aligns with distillation principles.
2. Appendix B discuss the  Correctness of RAPID’s Residual Distribution, proving RAPID’s sampling maintains the target distribution.

---

> ### Author Rebuttal · Authors · 2025-04-01
>
> Dear Reviewer SMJY,
>
> We sincerely appreciate your thorough review of our paper. Your constructive feedback will help us strengthen this work. Below are our responses to your concerns:
>
> ---
>
> ### 1. Missed References.
>
> A1: Thanks for pointing it out. REST[1] proposed selecting possible continuation by retrieving from a built datastore rather than generating with a draft LLM. TRIFORCE[2] introduced KV Cache compression for draft LLM based on chunk-wise similarity, which we have discussed in the related work and compared with in the response to Reviewer KbZ7. Ouroboros[3] sought to produce longer and more acceptable candidates from draft LLM per step based on draft phrases. Speculative RAG [4] did not utilize speculative decoding, but proposed a parallel draft-then-verify mechanism to improve RAG quality. **These works may be conceptually related to our work, but our method is quite distinct and orthogonal to them.** We will discuss the mentioned works (and further) in a revised version.
>
> [1] REST
>
> [2] TRIFORCE
>
> [3] Ouroboros
>
> [4] Speculative RAG
>
> ---
>
> ### 2. Overhead of each component.
>
> A2: Unlike regular RAG pipeline, which builds indexes for a large external corpus (hundreds of millions of documents), we only index/retrieve the chunks for the input long context (<128K) on-the-fly during inference. Therefore, **the RAG component latency in our method will become marginal compared to the inference latency over long context.** We now list the latency of each component of our RAPID on LongBench v2 (Long, CoT) with LLaMA-3.1-8B below:
>
> |  | RAG pipeline time (avg) | Prefix Time (avg) | Generation Time (avg) |
> | --- | --- | --- | --- |
> | LLaMA-3.1-8B-RAPID (Self-Spec) | 1.43 | 26.37 | 32.25 |
> | LLaMA-3.1-70B-RAPID (Self-Spec) | 1.43 | 163.43 | 121.76 |
>
> ---
>
> ### 3. Potential Bias of the RAG Drafter.
>
> A3: In section 4.5 of our paper, we have discussed the robustness of our RAPID to the retrieval quality. The results indicate that **our RAPID can maintain a stable (or better) performance even when the retrieval text is irrelevant to the long context**. This demonstrates the robustness of RAPID to retrieval quality, which is because the long-context target model in target distribution (Eq. 6) will preserve the long-context ability to verify candidates effectively, while RAG drafter provides benefits rather than reconstruction.
>
> ---
>
> ### 4. Explanation regarding upward speculation (using larger LLMs as drafter).
>
> A4: **For the rationality**, as the RAG based on compressed context introduces bias, even the superior LLMs may still lose some crucial information. As shown in Fig 2 of our paper, LLaMA-3.1-70B (RAG) on LongBench v2 achieves 23.66 points gain with another 13.72 points drop compared to LLaMA-3.1-8B (LC). **Despite the final gains of LLaMA-3.1-70B (RAG) over LLaMA-3.1-8B (LC) approach 10 points, there is still a large proportion of samples that cannot be solved by the superior LLaMA-3.1-70B (RAG).** However, our RAPID can integrate benefits from both long-context target LLM and stronger RAG drafter, **achieving performance improvements by incorporating the gains from drafter with minimum extra drops**.
>
> **For the efficiency**, larger RAG drafter will introduce more latency and require extra GPUs to serve. However, **our RAPID allows LLaMA-3.1-8B (LC) with LLaMA-3.1-70B drafter to achieve a comparable generation throughput to naive LLaMA-3.1-8B (LC)**. Empirically, the upward speculation of RAPID can serve as a turbo mode, which maximizes performance with comparable generation speed but consumes more resources. For a low-resource scenario, the self-speculation of RAPID can work well with both improved performance and efficiency and without additional resource requirments.
>
> ---
>
> ### 5. Use models from another family
>
> A5: Thanks for the insightful question. We now conduct a pilot experiment that uses LLaMA-3.1-8B-Instruct (LC) target LLM and Qwen2.5-7B-Instruct RAG drafter for RAPID. We convert the draft logit to target logit space and cut off the probability mass which is mismatched. The implementation is inspired by https://github.com/huggingface/transformers/blob/786d9c5ed920a099573ea7b6dbf265f1aeb32fc0/src/transformers/generation/candidate_generator.py#L783
>
> Surprisingly, using Qwen2.5-7B as RAG drafter can further improve the performance (though lower speedup due to logit alignment overhead). The gains may be due to the better short-context abilities of Qwen2.5-7B compared to LLaMA-3.1-8B upon RAG context. This indicates the essentiality of LLM capability in RAPID and opens more potential exploration of LLM combinations. Thanks for the constructive discussion.
>
> | Target | Draft | Overall | Overall (CoT) | Speedup |
> | --- | --- | --- | --- | --- |
> | LLaMA-3.1-8B | LLaMA-3.1-8B | 32.4 | 34.2 | 2.10 |
> | LLaMA-3.1-8B | Qwen2.5-7B | 34.0 | 34.8 | 1.81 |
>
> ---
>
> We hope the responses above can address your concerns and contribute to a reconsideration of review score.
>
> Best,
>
> Authors

---

> > ### Comment · Reviewer_SMJY · 2025-04-02
> >
> > Thank you for your response. I will adjust my rating to "accept" accordingly.
> >
> > All of my concerns have been addressed effectively, particularly regarding the acceleration between different model family.
> >
> > It seems that despite the associated costs, this area still holds potential for further exploration. If models from different families could be used for speculative decoding it would allow for better integration of their respective strengths while enabling the rapid adoption of the latest and most powerful models, even if they come from different vendors.
> > I’d like to hear author's thoughts on this issue.

---

> > > ### Author Response · Authors · 2025-04-08
> > >
> > > Dear Reviewer,
> > >
> > > Thank you for raising your rating and for your thoughtful feedback! Your comments have inspired us to recognize the significant potential for integrating more advanced LLMs with RAPID beyond just "long context" and "RAG".
> > >
> > > We believe the integration of two LLMs with RAPID would be substantially more efficient than directly ensembling outputs, as speculation significantly reduces generation latency for the target LLM. Furthermore, we see opportunities to combine multiple LLMs in a cascade fashion, where model B serves as a drafter for model A, while model C drafts for model B, and so on. This approach could effectively integrate the strengths of various LLMs in a computationally efficient manner. BTW, the alignment "tax" for multiple cross-family LLMs is also needed to consider, which will also be a valuable research direction for speculative decoding.
> > >
> > > We believe these intuitions from our discussion are highly promising and plan to explore these possibilities in our future work. We sincerely appreciate your engagement and have enjoyed our discussion.
> > >
> > > Best regards,
> > >
> > > Authors

---

### Official Review · Reviewer_s5Em · 2025-03-13

**Overall Recommendation:** 4

**Summary:**

The paper presents a novel decoding method called RAPID, designed to enhance the efficiency and quality of long-context inference in large language models (LLMs). RAPID introduces the RAG drafter—a draft LLM operating on shortened retrieval contexts—to speculate on the generation of long context target LLMs. RAPID operates in two settings: self-speculation, where the RAG drafter matches the target LLM's scale, and upward-speculation, where a larger RAG drafter assists a smaller target LLM. Both settings demonstrate effectiveness in improving performance and efficiency.

Main Results
1. RAPID achieves consistent performance improvements across different model scales and tasks. For example, LLaMA-3.1-8B with RAPID shows a performance increase from 39.33 to 42.83 on InfiniteBench.
2. RAPID provides significant speedup over long-context target LLMs, with up to 2.69× speedup for LLaMA-3.1-70B.
3.  RAPID enables effective knowledge transfer from larger RAG drafters to smaller target LLMs, further boosting performance. For instance, LLaMA-3.1-8B with a 70B RAG drafter achieves a performance of 49.98 on InfiniteBench.

## update after rebuttal
My primary concern was regarding the resource consumption and robustness of RAPID. The author's rebuttal has alleviated my concerns, so I will increase my rating from 3 to 4.

**Claims And Evidence:**

Yes

**Essential References Not Discussed:**

1. [REST: Retrieval-Based Speculative Decoding](https://aclanthology.org/2024.naacl-long.88/) (He et al., NAACL 2024)

**Experimental Designs Or Analyses:**

In conclusion, the experimental designs and analyses in the paper are sound and valid. They appropriately address the research questions, use relevant benchmarks and metrics, and provide comprehensive evaluations that support the claims made about RAPID's effectiveness for long-context inference in LLMs.

**Methods And Evaluation Criteria:**

Yes

**Other Comments Or Suggestions:**

No

**Other Strengths And Weaknesses:**

Strengths:
1. The paper presents a novel integration of speculative decoding with retrieval-augmented generation (RAG), creating a new paradigm for efficient long-context inference
2. The paper demonstrates significant speedups (over 2× in self-speculation settings) while maintaining or improving generation quality.
3. The method's effectiveness across different model scales (from 8B to 72B parameters) and diverse benchmarks suggests that it can be widely applied to various LLM architectures and tasks, enhancing its practical significance.
4. The paper is organized logically, with clear explanations of the methodology, experimental setup, and results.

Weaknesses:
1. The paper does not introduce entirely new theoretical frameworks but rather combines existing ideas in a way.
2. The upward-speculation setting requires additional computational resources (extra GPUs) to serve the larger RAG drafter.
3. Although the paper demonstrates robustness to suboptimal retrieval contexts, the method's performance can still be influenced by the quality of retrieval

**Questions For Authors:**

See Other Comments Or Suggestions

**Relation To Broader Scientific Literature:**

Yes

**Theoretical Claims:**

Yes.
1. Proof of Theorem 1 (Gradient of Distillation Loss)  The application of the chain rule, expansion of the log probability, and simplification using the Kronecker delta appear correct.
2. Proof of Correctness for RAPID's Residual Distribution. The proof logically demonstrates that the residual distribution ensures the overall sampling process still follows the target distribution.

---

> ### Author Rebuttal · Authors · 2025-04-01
>
> Dear Reviewer s5Em,
>
> We sincerely appreciate your thorough review of our paper. Your constructive feedback will help us strengthen this work. Below are our responses to your concerns:
>
> ---
>
> ### 1. The paper does not introduce entirely new theoretical frameworks but rather combines existing ideas in a way.
>
> A1: We believe big ideas are always composed of many small and existing ideas. While individual components (speculative decoding, RAG, long-context LLMs) may be familiar,  our contribution lies in the novel integration of RAG drafter for long-context target models using speculative decoding, with theoretically guaranteed inference-time transfer to combine both benefits from long-context LLMs and RAG. Our method is not an extension of any previous works, but provides a new perspective and solution for a long-time debate regarding “long-context LLMs or RAG”, which is meaningful for the application of long-context LLMs.
>
> ---
>
> ### 2. The upward-speculation setting requires additional computational resources (extra GPUs) to serve the larger RAG drafter.
>
> A2: We agree and especially point out that a larger RAG drafter will introduce more latency and require extra GPUs to serve. However, our RAPID can operate on two modes: (1) self-speculation for low-resource scenarios and (2) upward-speculation for high-resource scenarios. For a low-resource scenario, the self-speculation of RAPID can work well with both improved performance and efficiency and without additional resource requirements, which demonstrates great potential in real applications.
>
> While we believe **the upward speculation of RAPID can serve as a turbo mode, which maximizes performance with comparable generation speed, but consumes more resources.**
> For example, our RAPID enables LLaMA-3.1-8B (LC) with LLaMA-3.1-70B drafter to achieve 10 points accuracy gains with a comparable generation throughput to naive LLaMA-3.1-8B (LC), which allows us to explore the higher-bound performance wth a similar level of speed but more GPUs.
>
> ---
>
> ### 3. Although the paper demonstrates robustness to suboptimal retrieval contexts, the method's performance can still be influenced by the quality of retrieval.
>
> A3: We agree that RAPID’s performance can still be influenced by the quality of retrieval.  We are not only demonstrating the robustness of RAPID to retrieval quality in Section 4.5, but also seek to highlight that **our inference-time knowledge transfer will not hinder the target LLM from utilizing its long-context capabilities to reject low-quality candidates from RAG drafter as long as $\eta$ is properly set.** Our adjusted resampling distribution in Eq. (10) for rejected candidates guarantees that **the resampled tokens follow the exact distribution as direct sampling from the target model.** In other words, our RAPID can guarantee the lower-bound performance at the level of the target LLM as long as $\eta$ is properly set, while the retrieval quality only affects the performance “gains” and will not introduce too many drops.
>
> ---
>
> ### 4. Missed Reference.
>
> A4: Thanks for pointing it out. REST[1] proposed directly selecting possible continuation by retrieving from a built datastore rather than generating with a draft LLM. This work should be conceptually related to our work, but the method is distinct and orthogonal. We will discuss the mentioned works (and further) in a revised version.
>
> [1]  [REST: Retrieval-Based Speculative Decoding](https://aclanthology.org/2024.naacl-long.88/) (He et al., NAACL 2024)
>
> ---
>
> We hope the responses above can address your concerns and contribute to a reconsideration of review score. We also appreciate your careful review to point out missed references, which will be fixed in a revised version. Looking forward to discussing more with you.
>
> Best,
>
> Authors

---

> > ### Comment · Reviewer_s5Em · 2025-04-07
> >
> > Thank you for author response. I have no further questions and will increase my rating from 3 to 4.

---

> > > ### Author Response · Authors · 2025-04-08
> > >
> > > Dear Reviewer,
> > >
> > > Thank you for taking the time to review our rebuttal and for raising your rating. We appreciate your thoughtful consideration of our work and explanations. Your feedback has been valuable in helping us improve our paper.
> > >
> > > Best regards,
> > >
> > > Authors

---

### Official Review · Reviewer_bhTe · 2025-03-23

**Overall Recommendation:** 3

**Summary:**

This paper introduces Retrieval-Augmented Speculative Decoding (RAPID) that aims at both accelerating and enhancing generation quality in long-context inference. SD becomes inefficient with long contexts since both draft and target LLMs need to process complete context in memory. The authors introduce a RAG drafter, a draft LLM that operates on shortened retrieval contexts to speculate on the generation of long-context target LLMs. This method allows same-scale or even larger LLMs to function as drafters while maintaining computational efficiency. Additionally, the approach incorporates inference-time knowledge transfer, enabling stronger RAG drafters to enhance the final output quality.

**Claims And Evidence:**

**Claim:**  RAPID as an effective decoding method for accelerating long-context inference and, at the same time, enhancing generation quality through retrieval-augmented speculation

**Evidence:** In self-speculation settings (draft LLM of the same size as target LLM), RAPID achieves consistent performance improvements (e.g., 42.83 vs 39.33 on InfiniteBench for LLaMA-3.1-8B) with significant speedup (up to 2.69×) over the long-context target LLMs. The upward-speculation setting (draft LLM bigger than target LLM) further boosts performance improving LLaMA-3.1-8B from 42.83 to 49.98 on InfiniteBench), with comparable efficiency with the smaller long-context target LLMs.

**Essential References Not Discussed:**

Please see weakness

**Experimental Designs Or Analyses:**

RAPID is compared with four baselines (1) target LLM, (2) RAG, where the target LLM generates responses upon retrieval context of draft LLM input in RAPID, (3) naive Speculative Decoding (SD), which involves identical target and draft LLMs with RAPID but using the naive long-context target distribution, (4) MagicDec, with KV cache compression of draft model.

**Methods And Evaluation Criteria:**

RAPID is evaluated using LLaMA-3.1 (8B, 70B) and Qwen2.5 (7B, 72B) as target LLMs. The authors implemented two speculation settings: (1) self-speculation, where the RAG drafter matches the target LLM’s scale, and (2) upward-speculation, where RAG drafter is larger than the target LLM. For smaller models (LLaMA-3.1-8B, Qwen2.5-7B), they evaluate both settings, while larger models (LLaMA-3.1-70B, Qwen2.5-72B) they use self-speculation only.

Evaluation is performed on  $\inf$Bench and  LongBench v2. Efficiency metrics include: (1) prefill time and (2) speedup.

**Other Comments Or Suggestions:**

NA

**Other Strengths And Weaknesses:**

**Strengths:**
1. The paper is well-written and includes extensive experimental evaluation.

2. The results demonstrate improved performance with enhanced efficiency for long context LLMs.

**Weakness:**
1. In Figure 1, there is a sudden drop in accuracy for 32K tokens. Is there any possible explanation for this?

2. Generating the retrieval augmented target distribution requires computing target and draft distribution for each speculative token, which increases computational overhead. A FLOPs comparison with baseline should be provided for an estimate of the increased computation.

3. The prefill gets delayed compared to the baseline.

4. The generation throughput benefits should be clearly highlighted.

5. The prefill delay overhead to be discussed in details stating the delay associated to each component

6. The decode throughput: when it starts to become beneficial compared to the baseline SD, how it is a function of seq length

7. The memory and compute overhead to support the inference-time knowledge transfer should be discussed more.

**Questions For Authors:**

See before

**Relation To Broader Scientific Literature:**

The contributions should directly benefit the parallel verifier for SD.

**Theoretical Claims:**

1. For rejection, they sample from an adjusted residual distribution. This sampling strategy maintains theoretical guarantees. The authors prove in Appx. §B that the resulting tokens follow the same distribution as direct sampling from the original target model.

2. The gradient of the knowledge distillation loss L = T2 · KL(q(x)∥p(x)) with respect to the target LLM output is derived in Appendix A.

---

> ### Author Rebuttal · Authors · 2025-04-01
>
> Dear Reviewer bhTe,
>
> We sincerely appreciate your thorough review of our paper. Your constructive feedback will help us strengthen this work. Below are our responses to your concerns:
>
> ---
>
> ### 1. Possible explanation for the sudden drop in Figure 1.
>
> A1: This is an observed issue in RAG that performance may drop at certain lengths when more retrieval chunks are included [1]. The explanation from [1] is that the presence of certain hard negatives can mislead the LLMs and hinder their ability to generate accurate answers (even if relevant information is included). This issue cannot be directly addressed by using a stronger retrieval model. We believe the retrieval chunks between 16k and 64k may contain many hard negatives. Note that the drop in Fig 1 of our paper is not as “sudden” as displayed, as the x-axis is log-scaled.
>
> [1]Long-Context LLMs Meet RAG: Overcoming Challenges for Long Inputs in RAG
>
> ---
>
> ### 2. FLOPs comparison
>
> A2: Thanks for your suggestion. We now list the FLOPs of our RAPID and baselines per step (generating $\gamma$ tokens) below. As our RAPID consistently demonstrates higher acceptance rates than naive speculative decoding (SD) (shown in Fig. 3 of our paper), the FLOPs order will be Long Context >> SD > RAPID > RAG Drafter.
>
> |  | FLOPs |
> | --- | --- |
> | Long Context Target | $2\gamma T L + \gamma^2 T$ |
> | RAG Drafter | $2\gamma D L^{R} + \gamma^2 D$ |
> | SD | $\frac{2\gamma D L^{R} + \gamma^2 D+ 2T(L+ \gamma)} {\beta^{\text{SD}}}$ |
> | RAPID | $\frac{2\gamma D L^{R} + \gamma^2 D+ 2T(L+ \gamma)}{\beta^{\text{RAPID}}}$ |
>
>
>
> Target Model:
>
> - $T$ : Number of parameters in the target model.
> - $L$: Long context length.
>
> Draft Model:
>
> - $D$: Number of parameters in the draft model
> - $L^{R}$: The retrieval length of draft LLM input.
>
> Speculation:
>
> - $\gamma$: the generation length of draft model per step.
> - $\beta^{\text{SD}}$: the expectation of speculative decoding acceptance rate.
> - $\beta^{\text{RAPID}}$: the expectation of our RAPID acceptance rate.
>
> ---
>
> ### 3. The prefill gets delayed compared to the baseline
>
> A3: Yes, the speculative decoding mechanism does introduce extra prefill latency for the draft model. In our settings, however, the extra latency will be marginal (e.g, 26.37s vs 25.89s) as the draft input length is far shorter than the target one. Moreover, the two prefill stages are possible to be further optimized to overlap at the infrastructure level, which has the potential to totally erase the extra latency.
>
> ---
>
> ### 4. The generation throughput benefits should be clearly highlighted.
>
> A4:  We primarily list the generation throughput speedup in Table 1 and discuss it in Section 4.1. The efficiency benefits are also concluded in the abstract and introduction sections. We apologize for any confusion and are glad to strengthening the presentation of these efficiency benefits in revised version.
>
> ---
>
> ### 5. The prefill delay overhead should be discussed associated to each component.
>
> A5: The prefill delay overhead in Table 1 of our paper is only related to the prefill of the draft model upon the retrieval input. In addition, the RAG pipeline will introduce avg 1.43s latency per data sample. We will state the latency more clearly in a revised version.
>
> ---
>
> ### 6. The decode throughput: when it starts to become beneficial compared to the baseline SD, how it is a function of seq length.
>
> A6: In section 4.3 of our paper, we have discussed the impact of context and retrieval length for both performance and efficiency. The conclusions are (1) Our RAPID consistently improve the performance when the seq length > 8K. (When seq len < 8K, it is not really a long context and doesn’t need to apply RAPID) (2) Our RAPID starts to improve efficiency when the seq length is beyond 32k (with retrieval length < 16k). The longer the seq length, the more significant the speedup.
>
> ---
>
> ### 7. The memory and compute overhead to support the inference-time knowledge transfer.
>
> A7: As all required variables in Eq. (6) are necessary to compute for naive SD, **our RAPID would not introduce any extra memory overhead for the inference-time knowledge.** While the computation is just a tensor add operation, **the extra latency is also very marginal and can be ignored**.
>
> ---
>
> We hope the responses above can address your concerns and contribute to a reconsideration of review score. Looking forward to discussing more with you.
>
> Best,
>
> Authors

---

### Decision · Program_Chairs · 2025-05-01

**Decision:**

Accept (spotlight poster)

**Comment:**

The paper introduces RAPID, a novel integration of Retrieval-Augmented Generation (RAG) with Speculative Decoding (SD) for efficient long-context inference in LLMs. All reviewers found the problem timely and the method promising. The authors provided extensive empirical evidence on multiple model families and benchmarks, demonstrating improvements in both performance and efficiency. The theoretical justifications were also sound.

Several reviewers initially raised concerns about baselines, overhead, and novelty relative to prior work like TRIFORCE and REST. The authors provided a very thorough rebuttal, which addressed these concerns convincingly. All reviewers acknowledged the responses, and all four increased their scores, resulting in a consensus recommendation of "Accept".

Given the strong empirical results, clear methodological contributions, and effective rebuttal engagement, I believe this paper will be a valuable contribution to the conference and the broader community interested in long-context LLM efficiency.